# Case Study on the Impact of Water Resources in Beef Production: Corn vs. Triticale Silage in the Diet of Limousine × Podolian Young Bulls

**DOI:** 10.3390/ani13213355

**Published:** 2023-10-29

**Authors:** Carlo Cosentino, Rosanna Paolino, Francesco Adduci, Simona Tarricone, Corrado Pacelli, Emilio Sabia, Pierangelo Freschi

**Affiliations:** 1School of Agricultural, Forestry, Food and Environmental Sciences (SAFE), University of Basilicata, 85100 Potenza, Italy; carlo.cosentino@unibas.it (C.C.); sodextra.77@gmail.com (F.A.); corrado.pacelli@unibas.it (C.P.); emilio.sabia@unibas.it (E.S.); pierangelo.freschi@unibas.it (P.F.); 2Department of Soil, Plant and Food Science, University of Bari Aldo Moro, 70125 Bari, Italy

**Keywords:** Limousine × Podolian young bulls, beef production, feeding efficiency, water footprint

## Abstract

**Simple Summary:**

Agriculture accounts for 92% of the global freshwater footprint (WF), of which more than a quarter is used in livestock production for feed, mixing feed, watering animals and agricultural activities. This study shows that replacing maize silage with triticale silage in the diet of beef calves results in a relevant reduction in water consumption per cattle per day, without changing growth performance. It also shows how feed choice can help improve the water balance of livestock production, and thus reduce the pressure that the sector puts on water resources.

**Abstract:**

In this study, we have included the water footprint (WF) in the process of optimizing animal feed rations. The global footprint of cattle production accounts for the largest share (33%) of the global water footprint of livestock production. Using two homogeneous groups of Limousine × Podolian young bulls, two different diets were compared: corn silage feeding (CSF), with a corn silage-based diet; and triticale silage feeding (TSF), with a triticale silage-based diet. Silage constituted about 41% and 46% of the feed composition (for CSF and TSF, respectively). Diets were characterised by the same energy and protein content. Despite the lower WF in the TSF group than in the CSF group (7726 vs. 8571 L/day/calf respectively), no significant differences were found in animal performances (i.e., daily weight gain and final weight), feed conversion or income over feed costs. These results show that simple production decisions can have a significant impact on water resource. Therefore, the use of triticale silage should be further promoted, especially in world regions with limited water resources where low WF feed formulation is more strategic than elsewhere.

## 1. Introduction

Our dependence on water resources will increase significantly in the future, posing problems for future food security and environmental sustainability [1,2,3]. The European Green Deal and its Farm to Fork Strategy [4] aim to develop a sustainable food system along the whole value chain, from primary production to final consumption. Quantifying the water footprint (WF) of food consumption in the European Union and setting reduction targets are key topics of this strategy. The relationship between the freshwater resource and human productive activities, the Water Footprint Assessment (WFA), was developed to assess the amount of water consumed and water polluted. This concept was introduced by Hoekstra [5] and elaborated and validated by Chapagain and Hoekstra [6]. Cattle farming represents the largest share (33%) of the global livestock WF of production, followed by pigs (19%), dairy cows (19%) and poultry (11%), the latter which appears to be the most efficient sector in the use of natural resources, requiring 11 times less feed (in dry matter) than beef production [7,8,9]. Animal feeding, together with animal health and welfare assessment [10,11], plays a fundamental role in the economic and technical efficiency of animal production. Feeding accounts for about 60% of total costs in livestock farming and much of the water consumption in the livestock sector is used to produce feed [12,13,14,15,16,17]. Livestock feed production, which is a non-negligible cause of water pollution, takes up 70% of available agricultural land, including 33% of arable land and 8% of blue water used by humans [18]. Given the dual pressures of water scarcity and human nutrient needs, the basis for a sustainable supply of ruminant products is to clarify the consumption of water resources and the efficiency of water use in relation to the production of ruminant products, as well as the efficiency of nutrient conversion [19]. The use of irrigated maize has increased from 29% to 63% between 2004 and 2014 [20]. This has led to an increasing share of water resources being used for irrigation [21], steadily increasing the water footprint required for livestock feeding [22]. Compared to other cereals, triticale is more resistant to drought and disease, is suitable for low-input cultivation due to its low need for pesticides and can be grown on rather marginal land. With regard to the palatability and metabolizable energy of this grain, these aspects are mentioned as limiting factors for the use of triticale in the diet of monogastric animals only [23,24]. Against this background, the present study investigated the effect of replacing maize silage with triticale silage in the diet of Limousine × Podolian young bulls.

## 2. Materials and Methods

The trial was conducted on a farm in the Basilicata region, Italy, at an altitude of 600 m a.s.l. with forty 8-month-old Limousine × Podolian young bulls. Podolian cattle are an autochthonous breed belonging to the Hungarian Grey Steppe group and are reared in southern Italy [25,26], mainly in extensive management [27]. They are often crossed with specialized breeds to maximise meat production. During the experimental period, the animals, aged 230 ± 11 days, were kept in two different boxes with straw bedding (8.5 m^2^/head) in the same barn. Their initial average weight (mean ± SE) was 347.4 ± 0.741 kg in group 1 and 341.3 ± 0.636 kg in group 2. The manure management system used on the farm is liquid/slurry, where the manure is stored in the excreted form or with a minimal addition of water either in tanks or earthen ponds outside the barn, usually for a period of less than one year.

### 2.1. Diet Composition and Feeding

Two different diets were used: corn silage feed (CSF) for group 1 (*n* = 20) and triticale silage feed (TSF) for group 2 (*n* = 20). Triticale has a biological cycle that develops during the cold season (maize microtherm) and prefers high temperatures at the end of its cycle, therefore the WF is lower than maize, which shows an opposite behaviour in terms of heat and water requirements. The diets were formulated to be isoenergetic (0.90 UFV kg DM, 1 UFV 1820 kcal net energy) [28], with the same concentration of crude protein, crude fibre and starch as well as the same feed cost.

Feeding was administered using the total mixed ration (TMR) method [29,30] according to the composition given in Table 1. The TMR was sampled monthly and the chemical analysis of the TMR was performed according to the methods described in the scientific literature [31]. Using a NIRSYSTEM 5000 (Foss, Hillerød, Denmark) the following parameters were analysed: dry matter (DM), crude protein (CP), crude fibre (CF), neutral detergent fibre (NDF), acid detergent fibre (ADF), acid detergent lignin (ADL), ether extract (EE), ash and starch. The percentage of energy and PDI requirements of cattle were calculated according to the method proposed by Garcia et al. [32]. TMR was administered ad libitum to each group. Feed intake and feed refusal were measured every 14 days for each experimental group. There were no individual measurements of feed intake as the young bulls of each group were housed in the same box. The average feed intake for the group was calculated every 14 days according to the following relationship:Average daily feed intake (g/d) = (Total feed administered − Total feed refusal)/20

### 2.2. Live Weight and Daily Weight Gain

Live weight (LW) was measured every 14 days (approximately 6 h after administration of the daily ration) and average daily gain (ADG) in each period was calculated.

### 2.3. Feed Conversion Ratio and Income over Feed Cost

The assessment of the technical and economic feed rations given to each group during the experimental period was conducted by calculating, in each interval of 14 days, feed conversion ratio (FCR) and income over feed cost (IOFC). FCR is defined as consumed kg DM/kg LW produced and is used to evaluate the effects of feed quality, environment and management practices on production efficiency in cattle rearing and fattening [33]. The IOFC measures the difference between the production meat value and the feed cost and was calculated according to the following formula proposed by Bailey et al. [34]:IOFC = PLW × DWG − DFC
where PLW is the farm-gate price of calf live weight (EUR/kg), DWG is the daily weight gain (kg/d), and DFC is the daily feed cost (EUR/head).

### 2.4. Water Footprint Estimation

The WF of the live weight gain was calculated by adding feed WF (water for feed production), feed mix WF (water for feed mix), drinking WF (water intake) and service WF (water for cleaning the pen) according to the following formula [6,35]:WFA meat = WF feed + WF feed mixing + WF drinking + WF service

Green, blue and grey water were estimated during the experimental period for indirect and direct water footprint and live weight gain in kg. The green water footprint refers to soil moisture generated by evaporation of precipitation and used for crop production or moisture present in the product. The blue water footprint refers to evaporated surface or groundwater that enters the product or is reused elsewhere. The grey water footprint is defined as the amount of freshwater required to assimilate the pollutant load based on existing water quality standards [36]. Data from the literature were used to calculate the indirect water footprint of the feed used for both forages [7], while the other fractions (watering, mixing and service) were assessed on the farm using a mechanical water metre. The water used for mixing the animal feed was added to the blue water component in the feed ration.

### 2.5. Statistical Analysis

One-way ANOVA was applied to determine the effect of diet using R software (R Core Version 3.6.1, Vienna, Austria) [37]. Data are expressed as mean ± SE and differences between groups were tested by Student’s *t*-test.

## 3. Results and Discussions

The present study investigated the effects of replacing maize silage with triticale silage in the diet of Limousine × Podolian young bulls on animal performance and on the total water footprint required for livestock feeding.

No significant differences were found between the groups in the daily intake of DM. The intake of DM throughout the trial period was 8.00 kg/day for the SF group and 7.80 kg/day for the AF (Table 2).

The CSF and TSF groups achieved 596.43 and 585.91 kg LW and 1365 and 1341 kg/day DWG, respectively (Table 3). No significant differences were found between the two groups in terms of final LW and DWG.

Over the entire experimental period, FCR was 5.896 for the SF group and 5.857 for the AF group (Table 4).

Moeinoddini et al. [23] compared triticale- and corn-based diets in Holstein calves and found no effect on feed efficiency. In addition, their dietary treatment did not affect heart girth and body length. However, withers height and hip at weaning increased in calves fed triticale compared to the other diet. In the USA, Hill and Utley [38] compared three feedlot rations in finishing steers, consisting of corn only, corn/triticale and triticale only. The evaluation of steer performance and carcass quality traits showed no significant difference between treatment effects. The above-mentioned studies on calves are consistent with our research, which shows no differences in feed efficiency between triticale and corn feeds. In studies conducted on large numbers of animals, to understand functionality in beef production (e.g., muscularity and body condition score), predictive models like Legendre polynomials would be profitable [39].

The main concerns regarding the use of triticale in feed is due to the high potential for ergot contamination, which can have a negative impact on health and performance if, according to Shumann et al. [40], the growing bull feed contains up to 2.25 g of ergot or more than 400 µg of ergot alkaloid per kg DM. Other authors [41,42] point out that the focus should be on alkaloid concentration rather than ergot content, as the percentage of alkaloids in the different ergot sclerotia varies greatly [41,42].

Considering the whole experimental period, the income over feed cost (PLW = 3.50 EUR/kg LW; DFC = 0.420 EUR/kg DM × DM daily intake) was EUR 1.418/day in both groups over the whole experimental period (Table 5). Furthermore, the fixed costs associated with feed production and utilisation, such as silos, fencing, buildings and machinery, are an additional consideration when costing feed [43]. Given that feed costs represent such a large proportion of total costs, it is clear that effective management of feeding strategy decisions can contribute significantly to the economic sustainability and profitability of livestock farms [44].

The average water footprint of the ADG calculated over the whole experimental period was 6221.29 L in the CSF group and 5703.60 L in the TSF group. Therefore, the WF difference per kg LWG was 517.70 L between the two groups (Figure 1). The daily average intake of green, blue and grey water in the groups CSF and TSF was 76.09%, 13.59% and 10.32% and 82.45%, 6.37% and 11.18%, respectively. Gerbens-Leenes et al. [13], in a study on WF industrial beef production in four countries, reported values (L/kg LW) in the interval 4000–5000 in the NL and the USA, and close to 9000 in Brazil and 13,000 in China. This study also illustrated that choosing feed ingredients and sourcing wisely, and particularly substituting crops with co-products or crop residues, will help to improve the WP of livestock products, thus reducing the pressure the sector puts on scarce water resources. The highest total WF for beef production was evidenced in Brazil by Palhares et al. [45] with values ranging from 9249 to 23,521 L/kg LW. In another study in Arcadia Valley (MO, USA), Eady et al. [46] compared two rearing systems in beef cattle production, a farm with 634 cows delivering weaners and a farm with 720 cows delivering finished cattle; green water use ranged from 7400 to 12,700 L/kg LW depending on the class of livestock, with on-farm blue water use of 51–96 L/kg liveweight and off-farm blue water use of 0.1–59 L/kg LW.

The percentage of WF in the consumed feed components observed during the trial period was similar to the data reported by Chapagain and Hoekstra [35] for industrially bred cattle. As shown in Table 6, the percentage of WF feed was higher in the CSF group than in the TSF group (8471 vs. 7726), as there was a relevant water saving (745 L per bovine per day) in the TSF group.

According to Mekonnen and Hoekstra [7], feed consumption accounts for the largest share of water consumption in livestock production (98.83% and 98.70% in our study in TSF and AF, respectively), while the share of drinking, industrial and mixed water is quite low for both types of feed administration (<3%). Mourad et al. [47] observed a mean percentage of water allocation for WF of 98.3% in the central and northeastern region of Africa. Given the many variables involved in determining WF, it has been suggested by various authors [45,48] that general recommendations cannot be made on a large scale, e.g., to formulate policy recommendations, but only for individual operations. Furthermore, Broom [49], in a study that considered land use and conserved water data from different parts of the world, showed the large impact of farming systems on water resource use and the need to consider all systems when considering the impact of beef or other products on the global environment.

## 4. Conclusions

The sustainable management of agricultural and farming systems and tracking the impacts they cause is a complex task, and all the more so in a changing climate. The use of triticale silage in the diet instead of corn, when properly optimised, showed a lower water footprint in meat production, while maintaining the same technical and economic efficiency as feeding corn silage to cattle. These results confirm the possibility of including the water footprint parameter in ration optimisation and show that simple production decisions can have a significant impact on the consumption of water resources and on the sustainability of beef production. This study used data collected at the regional level, the use of which, rather than national average data for food ingredient production characteristics, provides a more accurate estimate of water resource impacts on beef production. One criticism of the present study could be that it was conducted for a single beef production system. The organization of primary data from individual studies into databases and expanding WF studies for the beef product will contribute to a better understanding of how water efficiency can be improved in this sector through a bottom-up approach.

## Figures and Tables

**Figure 1 animals-13-03355-f001:**
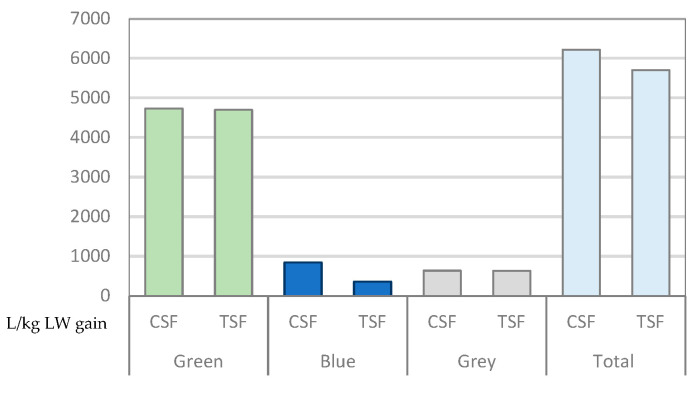
Water footprint (L/kg ADG) calculated over the entire experimental period. CSF: Corn silage feeding; TSF: Triticale silage feeding.

**Table 1 animals-13-03355-t001:** Composition, estimated nutritive values and costs of diet ^1^.

Components	CSF	TSF
	Diet composition, %	
Corn Silage	41.1	-
Triticale Silage	-	45.9
Corn Meal	13.7	13.7
Wheat Straw	13.7	6.9
Barley Meal	1.4	5.5
Corn Gluten Meal	-	3.8
Sunflower Meal	-	6.9
Soybean Meal Extraction	10.27	-
Beet Pressed Pulp	3.4	5.5
Corn Distillers	3.4	1.7
Hydrogenated Fat	1.0	-
Vitamin Mineral Supplement	1.4	1.4
NaHCO_3_	1.0	1.0
NaCl	0.7	1.03
Water Mixing	8.9	6.8
DM	58.4	58.3
	Feed cost	
EUR/kg DM	0.42	0.42
	Chemical composition, g/kg DM ^a^	
CP	147.4	147.0
CF	166.4	167.6
NDF	367.5	390.7
ADF	212.3	239.7
ADL	42.7	46.3
EE	43.0	27.1
Ash	78.9	87.9
Starch	248.8	249.7
	Nutritive value, kg/DM	
UFV ^b^	0.9	0.9
PDIN ^c^	96.5	106.0
PDIE ^d^	105.1	111.8
PDIA ^e^	51.5	59.5

^1^ CSF: Corn silage feeding; TSF: Triticale silage feeding; ^a^ Calculated by analysis of TMR; ^b^ UFV: Feed unit for meat production (net energy); ^c^ PDIN: Protein digested in the small intestine when rumen-fermentable nitrogen is limited; ^d^ PDIE: Protein digestible in the small intestine; ^e^ PDIA: Protein digestible in the small intestine supplied by rumen-undegraded dietary protein.

**Table 2 animals-13-03355-t002:** Average daily dry matter intake ($\bar{x}$ ± SE) ^1^.

Trial Day	CSF		TSF		*p-*Value
	DM	SE	DM	SE	
0	6.56	0.039	6.41	0.033	0.551
14	6.8	0.041	6.64	0.035	0.121
28	7.03	0.043	6.86	0.036	0.222
42	7.26	0.044	7.09	0.037	0.473
56	7.49	0.046	7.31	0.039	0.098
70	7.71	0.048	7.52	0.040	0.372
84	7.93	0.049	7.73	0.042	0.440
98	8.14	0.051	7.94	0.043	0.089
112	8.35	0.052	8.15	0.044	0.156
126	8.55	0.054	8.34	0.045	0.399
140	8.75	0.055	8.54	0.046	0.414
154	8.95	0.057	8.73	0.048	0.444
168	9.13	0.058	8.91	0.049	0.088
182	9.32	0.059	9.09	0.050	0.115
All	8.00	0.050	7.80	0.040	0.255

^1^ CSF: Corn silage feeding; TSF: Triticale silage feeding; DM: Dry matter.

**Table 3 animals-13-03355-t003:** Live weight (LW) and average daily gain (ADG) ($\bar{x}$ ± SE) ^1^.

Trial Day	LW, kg				*p-*Value	ADG, kg/day				*p-*Value
	CSF	SE	TSF	SE		CSF	SE	TSF	SE	
1	347.43	0.741	341.3	0.636	0.291	1.42	0.003	1.39	0.003	0.471
14	367.32	0.783	360.84	0.673	0.418	1.42	0.003	1.40	0.003	0.223
28	387.26	0.826	380.43	0.709	0.586	1.43	0.003	1.40	0.003	0.318
42	407.19	0.868	400.01	0.746	0.577	1.42	0.003	1.40	0.003	0.098
56	427.07	0.911	419.53	0.782	0.834	1.42	0.003	1.39	0.003	0.355
70	446.84	0.953	438.95	0.818	0.722	1.41	0.003	1.38	0.003	0.472
84	466.45	0.994	458.23	0.854	0.650	1.40	0.003	1.37	0.003	0.399
98	485.88	1.036	477.31	0.89	0.388	1.38	0.003	1.36	0.003	0.211
112	505.07	1.077	496.16	0.925	0.617	1.36	0.003	1.34	0.002	0.118
126	524	1.117	514.75	0.959	0.171	1.34	0.003	1.32	0.002	0.222
140	542.62	1.157	533.05	0.993	0.433	1.32	0.003	1.30	0.002	0.569
154	560.92	1.196	551.02	1.027	0.738	1.30	0.003	1.27	0.002	0.104
168	578.86	1.234	568.65	1.06	0.092	1.27	0.003	1.25	0.002	0.274
182	596.43	1.272	585.91	1.092	0.114	1.24	0.003	1.22	0.002	0.099
All	-	-	-	-	-	1.37	0.003	1.34	0.002	0.117

^1^ CSF: Corn silage feeding; TSF: Triticale silage feeding.

**Table 4 animals-13-03355-t004:** Feed conversion ratio (FCR) ($\bar{x}$ ± SE) **^1^**.

Trial Day	Feed Conversion Ratio				*p-*Value
	CSF	SE	TSF	SE	
0	4.629	0.08	4.603	0.09	0.233
14	4.777	0.05	4.748	0.03	0.455
28	4.935	0.09	4.904	0.02	0.128
42	5.104	0.11	5.072	0.12	0.241
56	5.285	0.07	5.251	0.06	0.364
70	5.477	0.06	5.442	0.05	0.455
84	5.682	0.07	5.645	0.05	0.451
98	5.900	0.07	5.860	0.04	0.624
112	6.131	0.06	6.089	0.09	0.131
126	6.376	0.09	6.332	0.03	0.832
140	6.635	0.05	6.589	0.09	0.221
154	6.910	0.07	6.861	0.07	0.119
168	7.200	0.08	7.149	0.05	0.417
182	7.507	0.07	7.454	0.09	0.151
All	5.896	0.05	5.857	0.09	0.331

^1^ CSF: Corn silage feeding; TSF: Triticale silage feeding.

**Table 5 animals-13-03355-t005:** Income over feed costs (IOFC) ($\bar{x}$ ± SE) **^1^**.

Trial Day	Income over Feed IOFC				*p-*Value
	CSF	SE	TSF	SE	
0	2.215	0.25	2.173	0.13	0.485
14	2.114	0.13	2.111	0.11	0.094
28	2.052	0.09	2.019	0.15	0.163
42	1.921	0.15	1.922	0.09	0.551
56	1.824	0.08	1.795	0.13	0.258
70	1.697	0.10	1.672	0.20	0.335
84	1.569	0.11	1.548	0.12	0.711
98	1.411	0.11	1.425	0.10	0.223
112	1.253	0.09	1.267	0.14	0.066
126	1.099	0.10	1.117	0.08	0.494
140	0.945	0.12	0.963	0.16	0.239
154	0.791	0.14	0.778	0.08	0.366
168	0.61	1.13	0.633	0.11	0.251
182	0.426	1.11	0.452	0.13	0.592
All	1.418	0.90	1.418	0.11	0.114

^1^ CSF: Corn silage feeding; TSF: Triticale silage feeding.

**Table 6 animals-13-03355-t006:** Trial period water footprint average (L/day/animal) ^1^.

Groups ^1^	Indirect Water Footprint		Direct Water Footprint		WF Average L/Day/Animal
	WF_Feed_	WF_Feed Mixing_	WF_Drinking_	WF_Service_	
	Estimated	Observed			
CSF	8471	1.439	23.99	75	8571
TSF	7626	1.756	23.41	75	7726

^1^ CSF: Corn silage feeding; TSF: Triticale silage feeding.

## Data Availability

Data are available on request.

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
