# Peer review of "Case Study on the Impact of Water Resources in Beef Production: Corn vs. Triticale Silage in the Diet of Limousine × Podolian Young Bulls"

_animals, 2023, doi:10.3390/ani13213355_

Round 1
Reviewer 1 Report
Comments and Suggestions for Authors
In this paper, TSF feed is used to conduct experiments to explore the influence of TSF feed on the water saving effect of beef cattle management. It was found that TSF feed did not affect daily weight gain, final weight, feed conversion and income over feed costs traits of cattle, but could reduce WF to some extent. The research method of this study is simple, and the data need to be supplemented. The conclusions are credible. However, the research direction is novel and compelling, so it is recommended to present the paper after extensive revisions.
Specifically, the following are the review comments:
1.The second paragraph of the introduction does not have a strong sense of logic and does not understand what is being said. It needs to be improved.
2. In lines 44-46, "Its production requires and pollutes large amounts of water, especially for the production of feed." Perhaps the sentence should be extended, No in-depth disclosure.
3. In line 48, "Water footprint is inversely related to feed conversion 48 efficiency." There is no reference, please add.
4. The introduction suggests deep thinking, looking at specific literature citations, rather than listing the literature without going into detail about the urgency and advantages of reducing water use in cattle.
5. In lines 94-96, the remarks can be displayed after the formula is in a column. It is recommended to delete "where".
6. Although there is no significant difference in the data in Table2, it is also necessary to increase the display of specific P-values. All tables should show p-values.
7. The result part is too simple, the discussion of the result and discussion together is not in-depth enough. It is suggested that the two parts be presented separately.
8. The Conclusions: Although TSF and CSF are used for daily weight gain, final weight, feed conversion and income over feed costs are no significant impact, but there is a lack of studies on other aspects, such as TSF's in-depth study on Meat quality or health. Is there any research data in relevant aspects suggesting adding or referring to other people's literature to prove one's opinion?
Author Response
Dear Reviewer,
We are grateful to you for your time and constructive comments on our manuscript. We have implemented your suggestions and wish to submit a satisfying revised version. Our answers to each comment are also reported below.
We look forward to hearing from you regarding our submission and to respond to any further question and comment you may have.
Thanks again for your last suggestions.
Sincerely yours
----------------------------
In this paper, TSF feed is used to conduct experiments to explore the influence of TSF feed on the water saving effect of beef cattle management. It was found that TSF feed did not affect daily weight gain, final weight, feed conversion and income over feed costs traits of cattle, but could reduce WF to some extent. The research method of this study is simple, and the data need to be supplemented. The conclusions are credible. However, the research direction is novel and compelling, so it is recommended to present the paper after extensive revisions. Specifically, the following are the review comments:
1.The second paragraph of the introduction does not have a strong sense of logic and does not understand what is being said. It needs to be improved.
- Thank you for your suggestion, we have reworded the paragraph accordingly
- In lines 44-46, "Its production requires and pollutes large amounts of water, especially for the production of feed." Perhaps the sentence should be extended, No in-depth disclosure.
- We have reworded the paragraph
- In line 48, "Water footprint is inversely related to feed conversion 48 efficiency." There is no reference, please add.
- We have added the reference
- The introduction suggests deep thinking, looking at specific literature citations, rather than listing the literature without going into detail about the urgency and advantages of reducing water use in cattle.
- We have tried to deepen the discussion in the introduction and we have added some references to better present the WF problem
- In lines 94-96, the remarks can be displayed after the formula is in a column. It is recommended to delete "where".
- Done
- Although there is no significant difference in the data in Table2, it is also necessary to increase the display of specific P-values. All tables should show p-values.
- Done
- The result part is too simple, the discussion of the result and discussion together is not in-depth enough. It is suggested that the two parts be presented separately.
- The chapter has been heavily reworded. Since we wanted to present the study in the ‘form of a 'communication', it is usual for this type of paper to combine results and discussion.
- The Conclusions:Although TSF and CSF are used for daily weight gain, final weight, feed conversion and income over feed costs are no significant impact, but there is a lack of studies on other aspects, such as TSF's in-depth study on Meat quality or health. Is there any research data in relevant aspects suggesting adding or referring to other people's literature to prove one's opinion?
- We have reworded the paragraph

Reviewer 2 Report
Comments and Suggestions for Authors
The paper, titled “Case study on the impact of water resources in beef production: corn vs. triticale silage in the diet of Limousine x Podolian young bulls“ addresses an important and timely topic. I found the subject matter of the article fascinating and read the manuscript with great interest. The paper aligns well with the scope of the journal. However, I believe that in its current form, it has several shortcomings.
The aim of this study is to explore the impact of dietary choices in cattle feed on water consumption and performance in beef calves. By replacing maize silage with triticale silage, the research demonstrates a substantial reduction in water use per cattle without compromising growth performance. This innovative approach of considering the water footprint in optimizing animal feed rations is a significant contribution, particularly as livestock production accounts for a substantial portion of the global water footprint. The study's strength lies in its practical implications for improving the water efficiency of cattle farming while maintaining animal performance, making it relevant for regions with limited water resources.
The paper generally aligns with the scope of the journal as it explores an important aspect of animal science, specifically beef production and its environmental impact, with a focus on water usage. However, it could be strengthened by more explicitly connecting the study's findings to the broader implications for animal science and sustainable agriculture, which would further enhance its alignment with the journal's scope. Additionally, considering the relevance of the study's findings to potential policy or practical applications in the field of animal science would be beneficial.
The main question addressed by the research is whether replacing maize silage with triticale silage in the diet of beef calves can reduce water consumption without negatively affecting growth performance.
The topic is relevant in the field as it addresses the significant issue of water usage in livestock production, specifically in beef calves. It explores a practical approach to potentially reducing the water footprint of beef production, which is an important consideration in sustainable agriculture. The study fills a gap in the field by examining the impact of feed choices on water usage in beef production.
This research adds valuable insights to the subject area by demonstrating that replacing maize silage with triticale silage can indeed reduce water consumption per cattle without compromising growth performance. It provides empirical evidence of a sustainable feed choice that can positively impact the water balance in livestock production.
Could the authors consider providing a concise overview in the introduction of additional challenges that impact beef production, beyond water supply, such as antibiotic usage? This would help to contextualize the broader issues in the industry. Please see 10.1016/j.rvsc.2023.03.008.
It would be valuable for the authors to include in the methods section a discussion of dietary analysis methods utilized by other researchers. This comparative analysis can offer readers a more comprehensive understanding of the methodologies employed in similar studies. Please see 10.3390/ani13050797.
It would be valuable to include an assessment of feed palatability in your study. Feed palatability is a critical factor influencing feed intake and, subsequently, animal performance. It can significantly affect the acceptance and consumption of specific feed components. Adding a section discussing feed palatability and citing relevant references in animal feeding practice would enhance the comprehensiveness of your study. Consider to cite: 10.1016/j.applanim.2020.105110
Did you assess the body condition score (BCS) of the cattle in your study? If so, please report the BCS results and provide an appropriate reference for the BCS assessment method you used. Consider citing: 10.1080/1828051X.2022.2032850.
The sample size and experimental design should be discussed in more detail. Providing information on the number of animals used, their characteristics, and the statistical methods employed would enhance the study's transparency and replicability.
The statistical methods described in the manuscript may benefit from further elaboration and clarity. It would be helpful to provide more details on the specific statistical tests employed, their rationale, and how they align with the research questions.
Could you please clarify whether you conducted tests for normality and homogeneity on your data before proceeding with the statistical analysis? It's crucial to ensure that the assumptions underlying your chosen statistical methods are met. I recommend referring to the guidelines outlined in [proposed reference, e.g., 10.1080/1828051X.2020.1827990] for conducting such tests to maintain the rigor and reliability of your analysis.
Explain how the data were presented and whether any transformations or adjustments were made to the raw data. Clarify how outliers, if any, were handled in the analysis.
To facilitate transparency and future research, consider sharing the data and detailed methodology used in this study.
Starting the discussion section by reiterating the aim of the study can provide clarity and context for readers.
While the study addresses the water usage aspect effectively, it would be valuable to briefly discuss potential economic implications of using triticale silage as an alternative feed. Beef production is a multifaceted industry, and economics play a vital role in decision-making.
The paper acknowledges the results, but it could be improved by discussing any potential limitations or weaknesses in the study design or findings. This would provide a more balanced perspective.
The implications of the research are significant for beef producers and the environment. By reducing the water footprint, it contributes to sustainable beef production. Discussing these practical implications in more detail would be beneficial.
The conclusions are consistent with the evidence and arguments presented in the paper. The study effectively demonstrates that replacing maize silage with triticale silage can reduce water consumption in beef calf production, which aligns with the main question posed.
The references appear to be appropriate and relevant to the research topic, providing necessary context and background information on water footprint and livestock production. However, the authors could further enrich the reference list by including more recent and wider-perspective papers in the field of sustainable beef production and water usage.
Please double-check the reference list to ensure that all references are included in the main text and vice versa.
In summary, the paper makes a valuable contribution to the field of sustainable beef production and water usage. Enhancements in the literature review, methodological clarity, economic considerations, and discussions of limitations and practical implications could further strengthen the research.
Author Response
Dear Reviewer,
We are grateful to you for your time and constructive comments on our manuscript. We have implemented your suggestions and wish to submit a satisfying revised version. Our answers to each comment are also reported below.
We look forward to hearing from you regarding our submission and to respond to any further question and comment you may have.
Thanks again for your last suggestions.
Sincerely yours
----------------------------
The paper, titled “Case study on the impact of water resources in beef production: corn vs. triticale silage in the diet of Limousine x Podolian young bulls“ addresses an important and timely topic. I found the subject matter of the article fascinating and read the manuscript with great interest. The paper aligns well with the scope of the journal. However, I believe that in its current form, it has several shortcomings.
The aim of this study is to explore the impact of dietary choices in cattle feed on water consumption and performance in beef calves. By replacing maize silage with triticale silage, the research demonstrates a substantial reduction in water use per cattle without compromising growth performance. This innovative approach of considering the water footprint in optimizing animal feed rations is a significant contribution, particularly as livestock production accounts for a substantial portion of the global water footprint. The study's strength lies in its practical implications for improving the water efficiency of cattle farming while maintaining animal performance, making it relevant for regions with limited water resources.
The paper generally aligns with the scope of the journal as it explores an important aspect of animal science, specifically beef production and its environmental impact, with a focus on water usage. However, it could be strengthened by more explicitly connecting the study's findings to the broader implications for animal science and sustainable agriculture, which would further enhance its alignment with the journal's scope. Additionally, considering the relevance of the study's findings to potential policy or practical applications in the field of animal science would be beneficial.
The main question addressed by the research is whether replacing maize silage with triticale silage in the diet of beef calves can reduce water consumption without negatively affecting growth performance.
The topic is relevant in the field as it addresses the significant issue of water usage in livestock production, specifically in beef calves. It explores a practical approach to potentially reducing the water footprint of beef production, which is an important consideration in sustainable agriculture. The study fills a gap in the field by examining the impact of feed choices on water usage in beef production.
This research adds valuable insights to the subject area by demonstrating that replacing maize silage with triticale silage can indeed reduce water consumption per cattle without compromising growth performance. It provides empirical evidence of a sustainable feed choice that can positively impact the water balance in livestock production.
- Dear reviewer, thank you for your comments and suggestions that we have accepted in order to improve the quality of our manuscript.
Could the authors consider providing a concise overview in the introduction of additional challenges that impact beef production, beyond water supply, such as antibiotic usage? This would help to contextualize the broader issues in the industry. Please see 10.1016/j.rvsc.2023.03.008.
- We have reworded the paragraph and added the suggested reference
It would be valuable for the authors to include in the methods section a discussion of dietary analysis methods utilized by other researchers. This comparative analysis can offer readers a more comprehensive understanding of the methodologies employed in similar studies. Please see 10.3390/ani13050797.
- We have added the suggested reference in M&M
It would be valuable to include an assessment of feed palatability in your study. Feed palatability is a critical factor influencing feed intake and, subsequently, animal performance. It can significantly affect the acceptance and consumption of specific feed components. Adding a section discussing feed palatability and citing relevant references in animal feeding practice would enhance the comprehensiveness of your study. Consider to cite: 10.1016/j.applanim.2020.105110
- We have added the suggested reference in M&M
Did you assess the body condition score (BCS) of the cattle in your study? If so, please report the BCS results and provide an appropriate reference for the BCS assessment method you used. Consider citing: 10.1080/1828051X.2022.2032850.
- We have added the suggested reference in Results and discussions
The sample size and experimental design should be discussed in more detail. Providing information on the number of animals used, their characteristics, and the statistical methods employed would enhance the study's transparency and replicability.
- We have reworded the paragraph
The statistical methods described in the manuscript may benefit from further elaboration and clarity. It would be helpful to provide more details on the specific statistical tests employed, their rationale, and how they align with the research questions.
- We have reworded the paragraph
Could you please clarify whether you conducted tests for normality and homogeneity on your data before proceeding with the statistical analysis? It's crucial to ensure that the assumptions underlying your chosen statistical methods are met. I recommend referring to the guidelines outlined in [proposed reference, e.g., 10.1080/1828051X.2020.1827990] for conducting such tests to maintain the rigor and reliability of your analysis.
- Statistical analysis was better specified
Explain how the data were presented and whether any transformations or adjustments were made to the raw data. Clarify how outliers, if any, were handled in the analysis.
To facilitate transparency and future research, consider sharing the data and detailed methodology used in this study.
Starting the discussion section by reiterating the aim of the study can provide clarity and context for readers.
- We have added the aim of the research into the results
While the study addresses the water usage aspect effectively, it would be valuable to briefly discuss potential economic implications of using triticale silage as an alternative feed. Beef production is a multifaceted industry, and economics play a vital role in decision-making.
- We have added a briefly discussion
The paper acknowledges the results, but it could be improved by discussing any potential limitations or weaknesses in the study design or findings. This would provide a more balanced perspective.
- We have added a comment into the results
The implications of the research are significant for beef producers and the environment. By reducing the water footprint, it contributes to sustainable beef production. Discussing these practical implications in more detail would be beneficial.
- We have added a comment into the conclusion
The conclusions are consistent with the evidence and arguments presented in the paper. The study effectively demonstrates that replacing maize silage with triticale silage can reduce water consumption in beef calf production, which aligns with the main question posed.
The references appear to be appropriate and relevant to the research topic, providing necessary context and background information on water footprint and livestock production. However, the authors could further enrich the reference list by including more recent and wider-perspective papers in the field of sustainable beef production and water usage.
Please double-check the reference list to ensure that all references are included in the main text and vice versa.
In summary, the paper makes a valuable contribution to the field of sustainable beef production and water usage. Enhancements in the literature review, methodological clarity, economic considerations, and discussions of limitations and practical implications could further strengthen the research.
- The multi-faceted aspects of the water footprint in livestock production and the additional challenges in beef production were highlighted. The bibliography was updated and the list of cited articles increased from 29 to 49.

Round 2
Reviewer 1 Report
Comments and Suggestions for Authors
This article, titled "Case study on the impact of water resources in beef production: 2 corn vs. triticale silage in the diet of Limousine Podolian 3 young bulls" discusses an urgent topic. I found the topic of this article very attractive and read the manuscript with great interest. This paper fits well with the scope of the journal.
Reviewer 2 Report
Comments and Suggestions for Authors
Dear Authors,
I wanted to extend my heartfelt congratulations to you and your team for the outstanding job you've done in revising your paper. I am genuinely impressed by the way you have meticulously incorporated the suggested revisions. Your commitment to improving the article's quality is evident, and I must say that the final result is nothing short of exceptional. The transformation from the initial draft to the current version is remarkable and a testament to your dedication to excellence.